# Molecular Lipopolysaccharide Di-Vaccine Protects from Shiga-Toxin Producing Epidemic Strains of *Escherichia coli* O157:H7 and O104:H4

**DOI:** 10.3390/vaccines10111854

**Published:** 2022-11-01

**Authors:** Ivan A. Dyatlov, Edward A. Svetoch, Anna A. Mironenko, Boris V. Eruslanov, Victoria V. Firstova, Nadezhda K. Fursova, Alexander L. Kovalchuk, Vyacheslav L. Lvov, Petr G. Aparin

**Affiliations:** 1State Research Center for Applied Microbiology and Biotechnology, Rospotrebnadzor, 142279 Obolensk, Moscow Region, Russia; 2National Research Center Institute of Immunology Federal Medical-Biology Agency (FMBA of Russia), 115522 Moscow, Russia; 3ATVD-TEAM Co., Ltd., 115522 Moscow, Russia

**Keywords:** STEC, clinically applicable LPS, antibody response, intraperitoneal infection, intragastric infection

## Abstract

Background: Shiga toxin-producing *Escherichia coli* (STEC) O157:H7 and O104:H4 strains are important causative agents of food-borne diseases such as hemorrhagic colitis and hemolytic–uremic syndrome, which is the leading cause of kidney failure and death in children under 5 years as well as in the elderly. Methods: the native *E. coli* O157:H7 and O104:H4 lipopolysaccharides (LPS) were partially deacylated under alkaline conditions to obtain apyrogenic S-LPS with domination of tri-acylated lipid A species—Ac_3_-S-LPS. Results: intraperitoneal immunization of BALB/c mice with Ac_3_-S-LPS antigens from *E. coli* O157:H7 and O104:H4 or combination thereof (di-vaccine) at single doses ranging from 25 to 250 µg induced high titers of serum O-specific IgG (mainly IgG1), protected animals against intraperitoneal challenge with lethal doses of homologous STEC strains (60–100% survival rate) and reduced the *E. coli* O157:H7 and O104:H4 intestinal colonization under an in vivo murine model (6–8-fold for monovalent Ac_3_-S-LPS and 10-fold for di-vaccine). Conclusions: Di-vaccine induced both systemic and intestinal anti-colonization immunity in mice simultaneously against two highly virulent human STEC strains. The possibility of creating a multivalent STEC vaccine based on safe Ac_3_-S-LPS seems to be especially promising due to a vast serotype diversity of pathogenic *E. coli*.

## 1. Introduction

*Escherichia coli* O157:H7 and O104:H4 are members of a group of pathogenic Shiga toxin-producing *E. coli* (STEC), which is defined by capacity to produce Shiga toxin (Stx) type 1 or 2, or both (as well as variants of these) [1,2,3]. Bacteria *E. coli* O157:H7 was first recognized as a human pathogen in the early 1980s [4,5,6], later it has been identified as the infectious agent causing worldwide morbidity and the largest STEC outbreaks [4,7,8,9,10,11,12,13,14,15,16,17,18,19]. *E. coli* O157:H7 causes various severe infections ranging from watery diarrhea to hemorrhagic colitis (HC), and Stx production determines the development of hemolytic uremic syndrome (HUS) [4,20,21], which is characterized by hemolytic anemia, thrombocytopenia, and renal failure [7,8,22]. Approximately 10% of patients with *E. coli* O157:H7 infection develop HUS complication with mortality rate up to 5% [7,22] and it is presently the most common cause of acute renal failure in children under 5 years and in the elderly [23,24].

The dangerous outbreak of food-borne infection in Germany and France (2011) caused by a relatively new highly virulent hybrid Stx type 2-producing *E. coli* O104:H4 strain [25] affected 3816 persons, 845 (22%) of treated patients developed HUS and among them 54 patients died [26]. Risk increase of HUS development in patients with HC and aggravation of the disease may be provoked by antibiotic therapy [27,28]. Today, 40 years after the first outbreak caused by STEC, an effective treatment for HUS is yet to be developed. Thus, the modern approaches to treatment of patients with HUS are unsuccessful and cannot prevent fatality. In view of this, the development of an effective vaccine against STEC is of particular relevance.

Main vaccine strategies against STEC infection are targeted on induction of anti-toxin and/or anti-whole cell immunity. Nontoxic subunits, recombinant proteins, and fusion proteins of various design developed on the basis of Stx molecule represent anti-toxin vaccine constructs, which induce strong humoral immune response as well as protection against STEC O157:H7 challenge [29,30,31].

Effective activation of immunity against bacterial cells is unlikely without induction of immune response to cell wall lipopolysaccharide (LPS). LPS O-polysaccharide domain, referred to as the O-antigen, determines the specificity of anti-STEC protective immunity [32,33].

LPS, the endotoxin of *E. coli*, along with Shiga toxins, is an important pathogenic factor, which participates in initiation of infection (invasion) as well as in the pathogenesis of HUS [16,34,35,36]. Numerous animal studies demonstrated that modelling of clinical symptoms of HUS is archived only with the combined administration of endotoxin (LPS) and Stx2 [37,38,39,40]. The importance of LPS in the pathogenesis of HUS provides a solid basis for the study of these macromolecules as target antigens for the development of vaccines against STEC. Additionally, high titers of serum antibodies against LPS of STEC pathogens are found in convalescent adult patients with HC and in children with acute HUS, which indicates the suppression of a high pathogenic activity of LPS [41,42,43]. Thus, LPS-specific immune response is advantageous during STEC infection and supports simultaneous formation of anti-toxin immunity, as well as immunity against cell wall antigens.

Endotoxicity, however, precludes LPS from clinical use. As LPS is highly heterogenous, we hypothesized that more homogenous pools of LPS might be less toxic. We developed a method to generate a homogenous LPS subfraction, Ac_3_-S-LPS, containing long chain O-specific polysaccharide (S-LPS) and mainly tri-acylated lipid A, with no penta- and hexa-acylated, and rare tetra-acylated lipid A [44]. Ac_3_-S-LPS had drastically reduced pyrogenicity and endotoxicity. In a clinical trial it was found that Ac_3_-S-LPS as a candidate vaccine against *S. flexneri* 2a was safe in volunteers at a dose of 50 µg, with low pyrogenicity, no major and few minor adverse events, and did not induce pro-inflammatory cytokines. In spite of the profound lipid A modification, the vaccine induced a prevalence of IgG and IgA antibodies [44].

In this manuscript, we report the development of a di-vaccine composed of two mono-vaccines based on the original molecular LPS form obtained from *E. coli* O157:H7 and O104:H4. Each LPS in di-vaccine is highly homogenous and represents Ac_3_-S-LPS molecules, in contrast to natural heterogeneous endotoxic LPS, which is a mixture of different pools of LPS molecules. Ac_3_-S-LPS from *E. coli* O157:H7 or O104:H4 is similar to clinically applicable Ac_3_-S-LPS vaccine from *S. flexneri* 2a and could be suitable for human parenteral administration.

## 2. Results

### 2.1. Structural Characterization of Ac_3_-S-LPS from STEC Strains

LPS was purified from *E. coli* O157 and O104 by the Westphal method [45]. S-LPS fraction with a long-chain O-polysaccharide domain was isolated from the intact *E. coli* O157:H7 and O104:H4 LPS by Sephadex G-150 gel-permeation chromatography in the presence of Na-deoxycholate, which was confirmed by electrophoresis in an SDS-polyacrylamide gel (Figure 1 and Appendix A).

S-LPS preparations were partially deacylated under alkaline conditions to obtain modified S-LPS. Lipid A (LA) was released from the modified S-LPS by mild acid hydrolysis. The negative ion mode electrospray ionization mass spectrum of the released LA showed peaks for (M-H)^−^ ions of various acyl variants (Figure 2). *E. coli* O157 and *E. coli* O104 modified S-LPS contain mainly tri-acylated LA with small admixtures of di- and tetra-acylated LA—Ac_3_-S-LPS. No peaks for penta- and hexa-acyl derivatives were present in the mass spectra of the modified LA.

^13^C NMR spectra of *E. coli* O157 and *E. coli* O104 Ac_3_-S-LPS (Figure 3) were in agreement with published data [46,47] indicating that deacylation did not alter the O-polysaccharide (O-PS) repeating unit structures. The O-PS domains of *E. coli* O157 and *E. coli* O104 Ac_3_-S-LPS contain unusual, rare sugars—α-D-perosamine and acetylneuraminic acid, respectively (Appendix A).

### 2.2. Immunobiological Study of Modified LPS

The serological specificity of the modified LPS was confirmed by a direct ELISA using rabbit sera obtained against *E. coli* O157 and *E. coli* O104 strains (data not shown).

Pyrogenicity of *E. coli* O157 and *E. coli* O104 Ac_3_-S-LPS was examined in rabbits by intravenous injection at a WHO approved Vi-vaccine pyrogenicity test dose of 0.025 μg/kg [48]. Both preparations were apyrogenic (∑Δ t °C = 0.4 °C for *E. coli* O157 Ac_3_-S-LPS and ∑Δ t °C = 0.3 °C for *E. coli* O104 Ac_3_-S-LPS).

The levels of endotoxin in intact LPS preparations from *E. coli* O157 and *E. coli* O104, estimated by the LAL assay, were high and constituted 584 and 3606 EU/µg, respectively. *E. coli* O157 Ac_3_-S-LPS had 29.86 EU/µg (≈19-fold reduction), and *E. coli* O104 Ac_3_-S-LPS had 33.24 EU/µg (≈108-fold reduction) of endotoxin content.

In the acute toxicity test, the dose of safe administration of the *E. coli* O157 and *E. coli* O104 Ac_3_-S-LPS exceeded 500 mg/kg per mouse after a single intraperitoneal (i.p.) injection.

### 2.3. Immunization Provides Effective Protection of Mice against Intraperitoneal STEC Infection

Intraperitoneal immunization of BALB/c mice with Ac_3_-S-LPS *E. coli* O157 or *E. coli* O104 at a repeated dose of 25 µg provided effective protection of animals against i.p. challenge with minimal lethal doses (LD_100_) of homologous STEC strains: 2.6 × 10^7^ CFU/mouse of *E. coli* O157 and 1 × 10^8^ CFU/mouse of *E. coli* O104 (Table 1). The preparation efficacy depended on the immunization scheme: the third injection provided 70% and 100% protection against *E. coli* O157 and O104 challenge, respectively, versus 60% protection rate for both strains following the second injection. All intact (control) mice died after i.p. injection of the same STEC strains at LD_100_, which were 2.6 × 10^7^ (*E. coli* O157) and 1 × 10^8^ (*E. coli* O104) CFU/mouse.

The protective properties of Ac_3_-S-LPS *E. coli* O157 or *E. coli* O104 correlated with their ability to elicit *E. coli* O157 or O104 LPS-specific IgG response, and both preparations were similar by immunogenic potency. Ac_3_-S-LPS *E. coli* O157 or *E. coli* O104 elicited serum O-specific IgG both after second and third injection at a dose of 25 µg (Figure 4A). The third injection of preparations elicited four-fold IgG increases over those following the second injection (the titers were 1:5120 versus 1:1280 for *E. coli* O157 Ac_3_-S-LPS and 1:10,240 versus 1:2560 for *E. coli* O104 Ac_3_-S-LPS), and these rises were statistically significant (*p* < 0.05, ANOVA). Thus, the well-known mechanism of serotype-specific anti-bacterial protection associated with rises in anti-O antibodies has been confirmed for modified LPS immunogens from STEC strains.

### 2.4. Protective and Immunogenic Properties of the STEC Di-Vaccine

The di-vaccine against *E. coli* O157 or O104 infection was obtained by mixing two antigens, Ac_3_-S-LPS *E. coli* O157 and Ac_3_-S-LPS *E. coli* O104, in a ratio of 1:1 (*w*/*w*). Both components were highly purified modified LPS with a minimal content of protein and nucleic acid impurities (less than 2% *w*/*w* each).

BALB/c mice were i.p. immunized with 25 µg of each modified LPS antigen (25 µg + 25 µg), i.e., the total dose of di-vaccine was 50 µg. The triple immunization with di-vaccine provided the highest protection (100%) against i.p. challenges with both highly virulent human STEC strains *E. coli* O157 or *E. coli* O104 (Table 1). In groups of mice immunized with di-vaccine and infected with *E. coli* O157 or *E. coli* O104, no bacteria were isolated from parenchymal organs 15 days after infection. These observations indicate the complete elimination of pathogens in vaccinated mice. Control mice died in all experiments.

In addition, the highest levels of serum IgG were elicited by immunization with di-vaccine preparation (Figure 4A): the end-point titer was 1:20,480 for both antigens after third injection, which was four-fold higher than IgG titer registered after the second injection–1:5120 for both antigens (*p* < 0.05, ANOVA). The IgG titers of control sera were less than 1:20 for each antigen. Thus, both Ac_3_-S-LPS components of di-vaccine induced 256-fold and higher increase in serum O-specific IgG-titers (*p* < 0.05, ANOVA). Notably, specific IgA in the serum and feces of immune mice was not detected under this protocol.

### 2.5. Non-Lethal Intragastric Colonization Study in Mice Immunized with E. coli (O157 + O104) Ac_3_-S-LPS Di-Vaccine and Its Individual Components

Bacterial counts in fecal samples of BALB/c mice double or triple immunized with a 25 µg dose of *E. coli* O157 and O104 Ac_3_-S-LPS determined on days 2, 5, 7, 9, and 12 after intragastric inoculation of 1 × 10^9^ CFU/mouse of homologous STEC strains resistant to streptomycin (*E. coli* O157^Str^ and *E. coli* O104^Str^) were not significantly different from those in the control group (data not shown). Therefore, a 25 µg dose of *E. coli* O157 and O104 Ac_3_-S-LPS provided no intestinal protective effect.

Due to the high safety profile of Ac_3_-S-LPS preparations, we decided to increase the experimental vaccination dose. After a 10-fold increase in a single immunizing dose of each Ac_3_-S-LPS antigen to 250 µg and to 500 µg for di-vaccine a significant decrease in bacterial counts in fecal samples of triple immunized mice was detected (Table 2) compared to the control group (*p* < 0.05, ANOVA). The highest decreases in *E. coli* O157 and *E. coli* O104 cell counts in the feces were observed in mice immunized with *E. coli* (O157 + O104) Ac_3_-S-LPS di-vaccine (*p* < 0.05, ANOVA). Reductions in bacterial counts in the feces of immunized mice were 8–9-fold for *E. coli* O157 and 5–7-fold for *E. coli* O104 in the early stages of the experiment on days 2 and 5 and reached 10-fold maximum on day 7.

### 2.6. Dose-Dependent Modification of the Systemic and Mucosal Immune Response with Ac_3_-S-LPS STEC Antigens

Of particular interest was the study of the immune response in mice immunized with a high dose of vaccine preparations and demonstrating mucosal immunity to STEC infection. *E. coli* O157 and O104 Ac_3_-S-LPS and *E. coli* (O157 + O104) Ac_3_-S-LPS di-vaccine elicited serum O-specific IgG after third injection at a dose of 250 µg of each antigen (Figure 4B). The highest levels of serum *E. coli* O157 and O104 LPS-specific IgG were again elicited by *E. coli* (O157 + O104) Ac_3_-S-LPS di-vaccine and the titer was 1:40,960 for both antigens. These were two- to four-fold higher (*p* < 0.05, ANOVA) than IgG titers following immunization with single-antigen preparations (1:10,240 for Ac_3_-S-LPS *E. coli* O157 or 1:20,480 for Ac_3_-S-LPS *E. coli* O104). The titers of control sera were <1:20 for each antigen, thus all Ac_3_-S-LPS preparations induced at least 512-fold rise in serum O-specific IgG-titers (*p* < 0.05, ANOVA). We found that IgG1 was the dominant subclass in the sera and feces (Figure 5A,B). The highest levels of IgG1 were elicited by Ac_3_-S-LPS *E. coli* (O157 + O104) di-vaccine and these titers for both antigens were 1:10,240 in the sera and 1:640 in the feces, which were 1024- and 64-fold higher (*p* < 0.05, ANOVA), respectively, than those in control sera and coprofiltrates (<1:10 for both antigens). Vaccine antigen Ac_3_-S-LPS *E. coli* O157 elicited 128- and 8-fold increases in serum or fecal IgG1 titers, respectively, compared to the control group (*p* < 0.05, ANOVA). Another antigen, Ac_3_-S-LPS *E. coli* O104, elicited 256 and 16-fold higher titers in the sera and feces, respectively, after triple injection, compared to the control group (*p* < 0.05, ANOVA). It should be noted that Ac_3_-S-LPS *E. coli* (O157 + O104) di-vaccine under the 10-fold dose escalation induced similar IgA response for both antigens in the sera (1:160 titer) and in the feces (1:80 titer) which were 16- and 8-fold higher (*p* < 0.05, ANOVA), respectively, than those for the control group (<1:10 for each antigen). Single Ac_3_-S-LPS *E. coli* O157 or Ac_3_-S-LPS *E. coli* O104 preparations were less immunogenic and elicited two- to four-fold increases in IgA titers in the sera and feces compared to the control group (*p* < 0.05, ANOVA). Modification of the serological response profile with increasing vaccine dose is an interesting feature of these antigens. Ac_3_-S-LPS immunogen becomes capable of the simultaneous activation of both the systemic and local specific immune responses.

## 3. Discussion

In this manuscript, we describe new antigens—modified lipopolysaccharides which were obtained from highly virulent human STEC strains *E. coli* O157:H7 and *E. coli* O104:H4. The choice of LPS as a protective antigen for the *E. coli* vaccine is not accidental. *E. coli* possess a plasticity in their genome that has allowed them to evolve into pathogenic strains able to cause clinically important diseases in humans [49]. In addition, genomic plasticity of *E. coli* is the reason for the wide antigenic variability of bacteria and the presence of many variations of both protein and polysaccharide antigens. Many *E. coli* antigens are bacterial proteins. Unlike them, LPS is a carbohydrate antigen, which is localized in the bacterial cell wall and can be considered as the most conserved *E. coli* antigen. According to the opinion of experts in the field of enteric infections, the development of a broad-spectrum *E. coli* vaccine is highly desirable. For that purpose, an interesting approach would be the use of immunologically cross-reactive polysaccharide antigens such as Poly-N-Acetylglucosamine (PNAG) [50]. Passive immunity to various strains of *E. coli* can be achieved by administration of immune sera from PNAG-injected animals. A vaccine consisting of PNAG conjugated with protein carrier has been developed [51,52].

Our approach aims to create a polyvalent *E. coli* vaccine based on clinically applicable 3-acyl derivatives of LPS. Partial deacylation of native LPS did not alter the O-polysaccharide repeating unit structures, but significantly reduced LAL clotting potency of Ac_3_-S-LPS *E. coli* O157 and Ac_3_-S-LPS *E. coli* O104 and both preparations met pyrogenicity requirements for a capsular polysaccharide vaccine [48]. This article describes a divalent LPS vaccine against the most epidemically significant *E. coli* strains O157:H7 and *E. coli* O104:H4. However, the LPS vaccine preparation also contains the cross-reactive core domain common to all *E. coli* strains belonging to the R2 type. This fact can serve as the basis for the formation of a wider spectrum of protection against various *E. coli* strains than O-serotype specific protective immunity.

We found that double and triple i.p. immunization with Ac_3_-S-LPS *E. coli* O157 or Ac_3_-S-LPS *E. coli* O104 or di-vaccine combination thereof at a dose of 25 µg of each antigen elicited *E. coli* O157 and O104 LPS-specific IgG response in mice, and IgG titers were 64-fold and higher than those for the control group. Thus, *E. coli* O157 and O104 LPS-specific IgG confer systemic protective immunity against i.p. challenge with homologous STEC strains at a lethal dose (60–100% protective efficacy), which agrees with similar results of other researchers [53,54,55]. However, protection against the oral administration of pathogenic *E. coli* O157 and *E. coli* O104 in mice vaccinated three times with LPS monocomponents, or di-vaccine, at doses of 25 or 50 μg, has not been achieved.

A number of researchers have questioned the suitability of parenteral immunization for the induction of intestinal immunity. Neither the *E. coli* O157 O-polysaccharide-based vaccine nor i.p. injection of *E. coli* O157-specific monoclonal antibodies protected animals against intragastric infection with *E. coli* O157:H7. *E. coli* O157-specific IgG was not detected in feces after subcutaneous immunization [56,57]. In this regard, interesting data were obtained when we increased the dose of LPS immunogens 10-fold. We found that triple i.p. immunization with 250 µg of Ac_3_-S-LPS *E. coli* O157 or Ac_3_-S-LPS *E. coli* O104 or di-vaccine based on antigen compounds at a dose of 500 µg induced local intestinal immunity in mice. Limitation of the colonization activity of *E. coli* O157:H7 and *E. coli* O104:H4 after intragastric infection with homologous STEC strains (at a non-lethal dose of ~1 × 10^9^ CFU/mouse) were registered in immune mice. The concentrations of pathogens in the feces of immunized mice during the entire study period (2, 5, 7, 9, and 12 days after infection) were significantly lower (up to 10-fold) than in the control group. The obtained protective effect is consistent with the study by Ademokoya et al. [58], which showed that i.p. immunization of rats with native *E. coli* O157 LPS equally effectively reduced the concentration of pathogen in the liver, spleen, ileum, and in the feces and provided 70% protection against intragastric infections with lethal doses of *E. coli* O157:H7 compared with 100% lethality in control mice.

The decrease in STEC pathogen’s CFU counts in the feces correlated with high levels of *E. coli* O157 and O104 LPS-specific IgG in the sera (512-fold and higher than those for the control group) and feces with domination of IgG1 subclass in mice triple i.p. immunized with 250 µg of Ac_3_-S-LPS *E. coli* O157 or Ac_3_-S-LPS *E. coli* O104 and di-vaccine combination thereof at a dose of 250 µg of each antigen. Moreover, *E. coli* O157 and O104 LPS-specific IgA were detected in immunized mice sera and feces. The detection of *E. coli* O157 and O104 LPS-specific IgG in the feces confirms the suggestions by Konadu et al. [32] regarding the possible transudation of IgG from the blood into the intestinal lumen. Our data are also in agreement with Robbins et al.’s [59] hypothesis, according to which anti-LPS IgG is a major immunoglobulin component of secretory fluids, including those of the small intestine, that may inactivate the inoculum of the pathogen on epithelial surfaces of the intestine. An important protective role of antibodies against STEC was demonstrated by Paton et al. [60], who showed that polyclonal antibodies against *E. coli* O111 and O157 blocked the adhesion of homologous strains to human intestinal epithelial cells by 95%.

Our findings demonstrate that parenteral (i.p.) immunization with *E. coli* (O157 + O104:H4) Ac_3_-S-LPS di-vaccine induces both systemic and intestinal anti-colonization immunity in mice simultaneously against two epidemically significant human STEC strains; and this protects against i.p. and intragastric infection with homologous STEC strains. The immunogenic and protective properties of Ac_3_-S-LPS di-vaccine *E. coli* (O157 + O104) were higher in comparison with single-antigen preparations. This phenomenon may be explained by the synergistic action of two antigenically different STEC Ac_3_-S-LPS on the immune system of mice, and absence of intermolecular antigenic competition. It is also plausible to propose that the higher level of protection is due to the induction of antibodies to the core domain.

The clinical safety level of the original LPS immunogens opens up the possibility of developing a polyvalent Ac_3_-S-LPS vaccine. These findings are the basis for our further studies of the immunogenic and protective properties of preparations containing Ac_3_-S-LPS antigens isolated from other STEC strains O26, O45, O103, O111, O121, and O145, causative agents of HUS.

## 4. Materials and Methods

### 4.1. Bacterial Strains and Growth Conditions

*E. coli* O157:H7 strain 717 51658 and *E. coli* O104:H4 strain RKI#112027 were provided by the State Collection of Pathogenic Microorganisms and Cell Cultures of the State Research Center for Applied Microbiology and Biotechnology (SRC AMB), Obolensk, Russia. Strains of *E. coli* O157:H7 and *E. coli* O104:H4 resistant to streptomycin—*E. coli* O157^Str^ and *E. coli* O104^Str^—were generated by spontaneous mutagenesis and used in this study both for intragastric and i.p. infection. Biomass of *E. coli* O157:H7 and *E. coli* O104:H4 were obtained through cultivation in SORBITOL nutrient medium (SRC AMB, Obolensk, Russia) in a 60 L BioR fermenter (Prointech, Saratov, Russia) for 18 h at 37 °C with stirring and aeration. Bacterial cells inactivated by 1.5% formaldehyde were separated from the liquid phase by flow centrifugation (Z-41, Cepa, Lahr, Germany). Wet cells were subsequently washed with filter-sterilized saline and distilled water, and resuspended in sterile water. Then 200 mL aliquots were loaded into 1200 mL cylindrical glass vials (Fast-Freeze, Labconco, Kansas City, MO, USA) and frozen in a freezer (Shell Freezer, Labconco, Kansas City, MO, USA) at −40 °C for 40 min. Then vials were transferred to a lyophilizer (FreeZone, Labconco, Kansas City, MO, USA) and freeze-dried at −50 °C for 12 h.

### 4.2. Preparation of S-LPS

LPS was obtained from *E. coli* O157 and O104 by the Westphal method [58]. The crude LPS was purified according to Ledov et al. [48]. Briefly, LPS was dissolved in 0.05 M TRIS-buffer and treated with RNAse, DNAse, and Proteinase K (Sigma-Aldrich, Burlington, MA, USA). The suspension was dialyzed using ultrafiltration with a 50 kDa cut-off membrane (Vladisart, Vladimir, Russia), concentrated, and lyophilized to give a purified LPS preparation. LPS was applied to a column of Sephadex G-150 in 0.2 M NaCl containing 0.001 M EDTA, 0.01 M TRIS, 0.01% NaN_3_, 2% Na-deoxycholate (Sigma-Aldrich, Burlington, MA, USA), and then eluted with the same buffer containing 0.25% Na-deoxycholate [61]. Fractions that contained S-LPS were combined and freeze-dried.

### 4.3. Partial Deacylation of S-LPS

A solution of S-LPS was heated with stirring in 8.3% aqueous ammonia and water (100 mL) containing 100 mg of Na-deoxycholate at 30 °C for 8 h, then cooled to 5–10 °C, diluted with sterile water, neutralized with AcOH, and freeze-dried. The product was treated with 100% ethanol, the precipitate was separated by centrifugation, washed twice with 100% ethanol, vacuum-dried, dissolved in sterile water and freeze-dried to give S-LPS with mainly a tri-acylated lipid A—Ac_3_-S-LPS moiety. Immediately before injection into animals, the freeze-dried Ac_3_-S-LPS preparations were suspended and diluted in sterile saline.

### 4.4. Analyses

Protein and nucleic acid contents in the LPS preparations were determined by the methods of Bradford [62] and Spirin [63], respectively. S-LPSs were analyzed using sodium dodecyl sulfate (SDS)-polyacrylamide gel (12%) electrophoresis with silver staining [64]. ^1^H and ^13^C nuclear magnetic resonance (NMR) spectroscopy of Ac_3_-S-LPS was performed for solutions in 99.95% D_2_O at 323K on a Bruker DRX-500 spectrometer (Bruker Daltonics, Billerica, MA, USA) using sodium 3-trimethylsilylpropanoate-2,2,3,3-d_4_ (δ_H_ 0) and acetone (δ_C_ 31.45) as references for calibration. Prior to analysis, samples were freeze-dried from 99.5% D_2_O. The Bruker Topspin 2.1 program was employed to acquire and process the NMR data. Electrospray ionization high-resolution mass spectra were recorded in the negative ion mode using a micrOTOF II mass spectrometer (Bruker Daltonics, Billerica, MA, USA). To obtain lipid A, Ac_3_-S-LPS was treated with 2% AcOH at 100 °C for 1 h. Samples (~50 ng μL^−1^) were dissolved in a 1:1 (*v*/*v*) water-acetonitrile mixture and sprayed at a flow rate of 3 μL min^−1^. End plate offset voltage was set to −0.5 kV and capillary voltage to −4.5 kV. Drying gas temperature was 180 °C. Mass range was from *m*/*z* 50 to 3000.

### 4.5. Safety Characteristics of Ac_3_-S-LPS

The level of endotoxin in LPS and Ac_3_-S-LPS was assayed by the *Limulus polyphemus* amebocyte lysate (LAL) method with FDA-licensed Endosafe^®^ chromogenic LAL reagent (Charles River, Wilmington, MA, USA), according to the manufacturer’s instructions. The results were expressed in endotoxin units (EU) per µg. Pyrogen tests were conducted with doses at 0.025 µg/kg on Chinchilla rabbits weighing 2.8–3.05 kg in accordance with the European Pharmacopoeia requirements [65] and WHO requirements for the Vi-polysaccharide vaccine [48]. A substance was considered to be apyrogenic if the cumulative temperature rises of three rabbits did not exceed 1.2 °C.

### 4.6. Mice

Pathogen-free 8-week-old female 20 to 22 g BALB/c mice were obtained from a colony at branch “Stolbovaya” of the Federal State Institution of Science “Scientific Center of Biomedical Technologies of FMBA”, Russia. All animal experiments were performed in accordance with protocol 5512 approved by the Institutional Animal Care and Use Committee of the NRC-Institute of Immunology, Moscow, Russia. This protocol is in compliance with the NIH guidelines for the humane use and care of laboratory animals.

### 4.7. Immunization

To investigate the immunogenicity and efficacy of *E. coli* O157 and *E. coli* O104 Ac_3_-S-LPS, BALB/c mice (15 mice per group) were immunized i.p. twice with 30-day interval or three times with 10-day intervals, respectively. Single immunizing doses of *E. coli* O157 and *E. coli* O104 Ac_3_-S-LPS and di-vaccine in different experiments ranged from 25 to 250 µg. Collection of blood and fecal samples was carried out 15 days after the final immunization. Control mice were i.p. injected with sterile saline.

### 4.8. Serology

LPS antibody levels in sera and feces from 5 mice were assayed by enzyme-linked immunosorbent assay (ELISA) using 96-well high-binding plates (Greiner, Sigma-Aldrich, Burlington, MA, USA) coated with native LPS of *E. coli* O157:H7 or *E. coli* O104:H4 (40 µg/mL) in PBS (Sigma-Aldrich, Burlington, MA, USA). Peroxidase-conjugated secondary antibodies specific for mouse IgG, IgG_1_, IgG_2_, and IgA (Invitrogen) were used for detection. The LPS antibody levels were expressed in end-point titers (optical density ≥ 0.4).

### 4.9. Intraperitoneal Challenge Model

To reproduce the lethal infection, mice (10 per group) were i.p. challenged with the minimum lethal dose of homologous STEC strains: 2.6 × 10^7^ CFU/mouse of *E. coli* O157 and 1 × 10^8^ CFU/mouse of *E. coli* O104. The infected mice were observed for two weeks. Surviving mice were euthanized with CO_2_. All animals, including the dead, were necropsied and their livers and spleens were removed, homogenized, and plated on GRM agar with streptomycin (100 mg/L) (Ecolan, Moscow, Russia). Plates were incubated at 37 °C for 18–24 h. Cultures of *E. coli* O157 and *E. coli* O104 were identified by latex agglutination using a homemade kit.

### 4.10. Non-Lethal Intragastric Colonization Study

To reproduce a non-lethal colonization model, BALB/c mice (10 per group) were immunized i.p. twice with a 30-day interval or three times with 10-day intervals, respectively. On day 45 mice, were intragastrically infected with *E. coli* O157^Str^ or O104^Str^ (1 × 10^9^ CFU/mouse) by the method proposed by Wadolkowski et al. [66]. Fecal bacterial counts (CFU/g) were determined by preparing serial ten-fold dilutions (up to 1:1,000,000 times) of homogenized stool samples in 0.8% NaCl (Sigma-Aldrich, Burlington, MA, USA) and plating them on GRM agar with streptomycin (100 mg/L) (Ecolan, Moscow, Russia).

### 4.11. Statistical Analyses

Statistical analyses were performed with STATISTICA 10.0 software (StatSoft, Tulsa, OK, USA). Antibody levels are expressed as geometric mean titer (GMT). Levels below the sensitivity of the ELISA were assigned the value of one-half of that level. Comparison of GMTs were performed by ANOVA. *p*-values less than 0.05 were considered significant.

## Figures and Tables

**Figure 1 vaccines-10-01854-f001:**
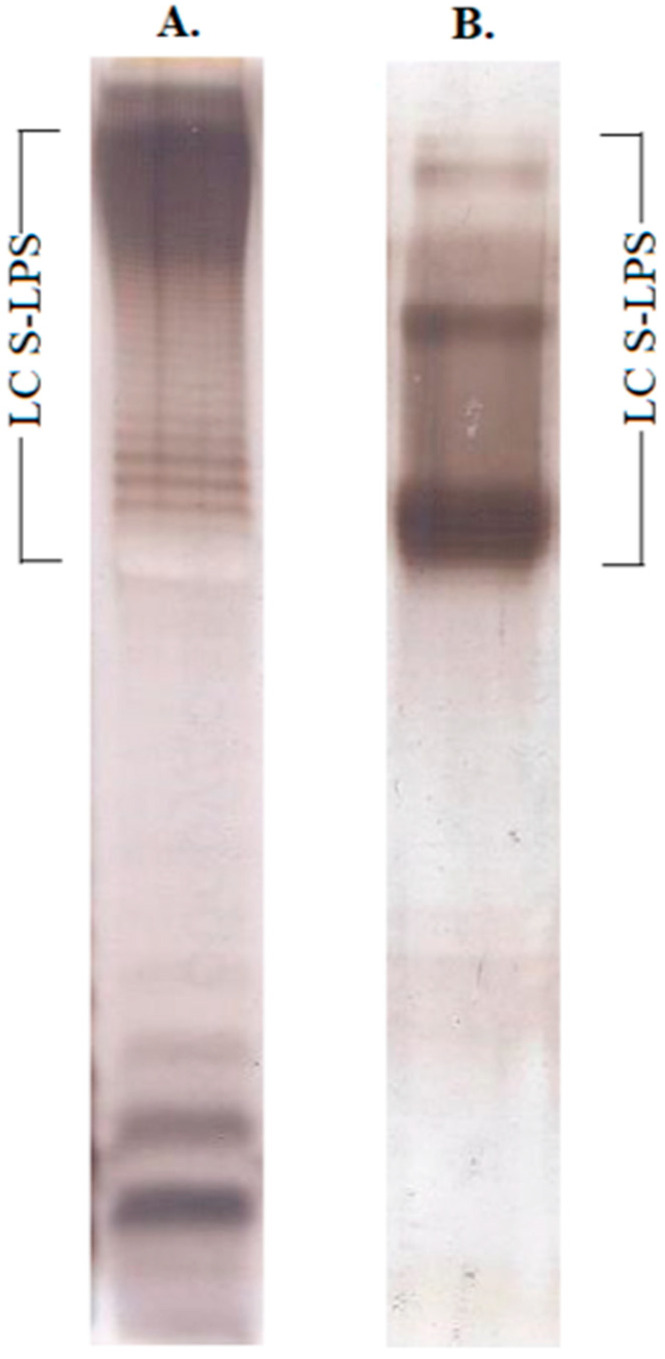
Silver-stained SDS polyacrylamide gel of S-LPS from *E. coli* O157 (**A**) and *E. coli* O104 (**B**). LC S-LPS indicate S-LPS with long-chain O-polysaccharide.

**Figure 2 vaccines-10-01854-f002:**
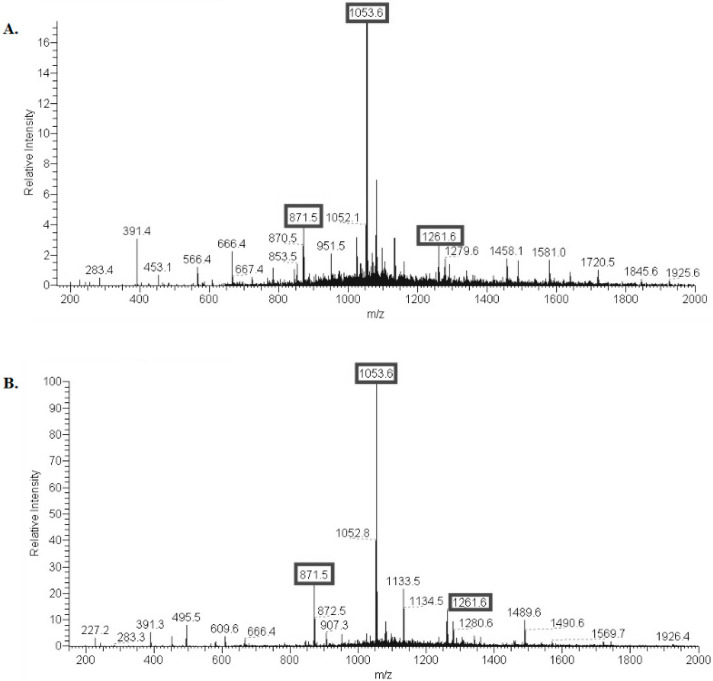
Mass spectra of lipid A (LA) of Ac_3_-S-LPS *E. coli* O157 (**A**) and of Ac_3_-S-LPS *E. coli* O104 (**B**). The peak at *m/z* 1053.6 (major) belongs to tri-acylated LA; peaks at *m/z* 871.5 and 1261.6 belong to di- and tetra-acylated LA, respectively. All species of LA are monophosphoryl.

**Figure 3 vaccines-10-01854-f003:**
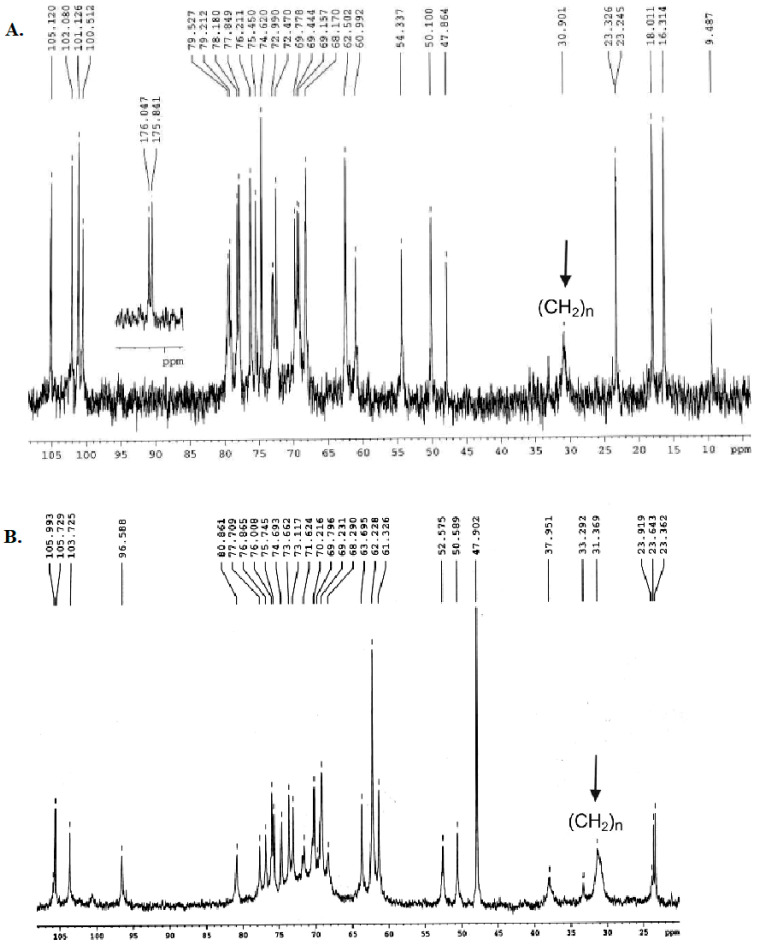
^13^C NMR spectra of Ac_3_-S-LPS *E. coli* O157 (**A**) and Ac_3_-S-LPS *E. coli* O104 (**B**). (CH_2_)_n_ indicates methylene groups of fatty acids.

**Figure 4 vaccines-10-01854-f004:**
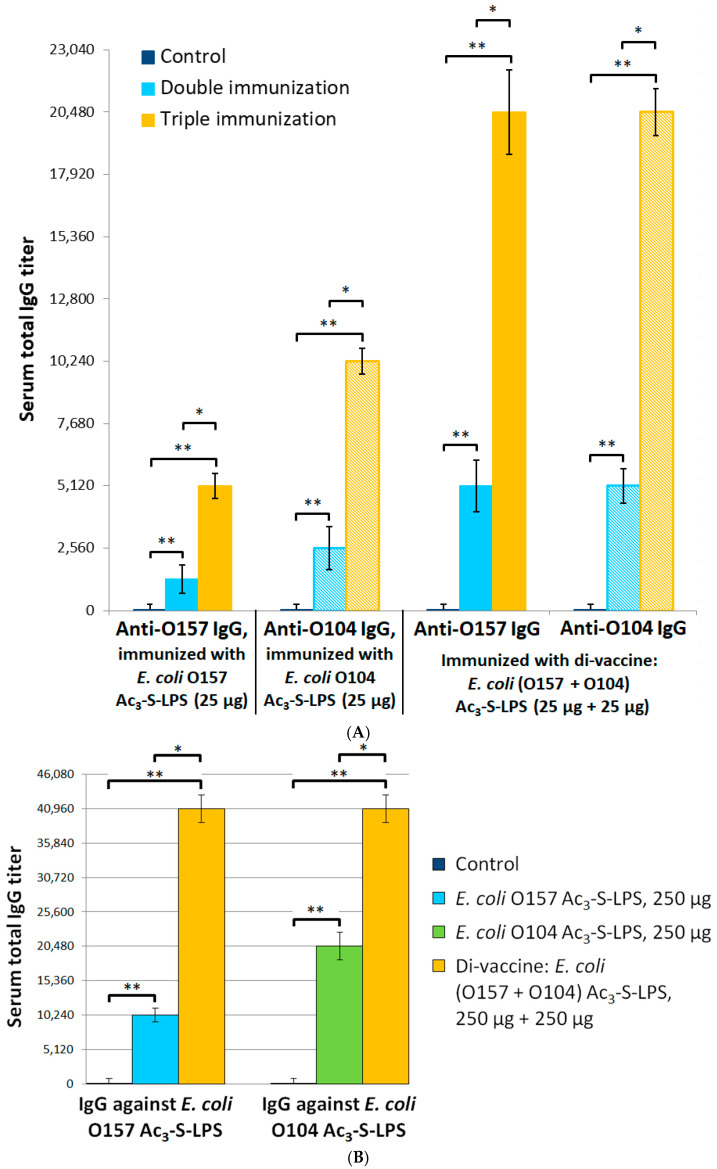
Total IgG serum antibody response in mice induced by *E. coli* (O157 + O104) Ac_3_-S-LPS di-vaccine and its individual components. (**A**) Mice were immunized i.p. two or three times with 25 µg of each Ac_3_-S-LPS *E. coli* O157 or Ac_3_-S-LPS *E. coli* O104 antigen or with 50 µg (25 µg + 25 µg) of di-vaccine. The third injection of preparations elicited increases in titers over those following the second injection, and these rises were significant (*p* < 0.05, ANOVA). The highest levels of serum total IgG were elicited by immunization with di-vaccine preparation. The titers of control sera did not exceed less than 1:20 for both antigens, and the increases in total IgG titers over those in control sera were highly significant (*p* < 0.01, ANOVA). (**B**) Total IgG serum antibody titers in mice immunized i.p three times with 250 µg of each Ac_3_-S-LPS *E. coli* O157 or Ac_3_-S-LPS *E. coli* O104 antigen or with 500 µg (250 µg + 250 µg) of di-vaccine. The di-vaccine elicited two- to four-fold IgG increases over those following the immunization with single-antigen preparations, and these rises were significant (*p* < 0.05, ANOVA). All Ac_3_-S-LPS preparations induced 512-fold and higher rises in serum O-specific IgG-titers and these rises were significant (*p* < 0.01, ANOVA). The results shown are means ± SD. Significant differences between values are indicated (* *p* < 0.05, ** *p* < 0.01, ANOVA).

**Figure 5 vaccines-10-01854-f005:**
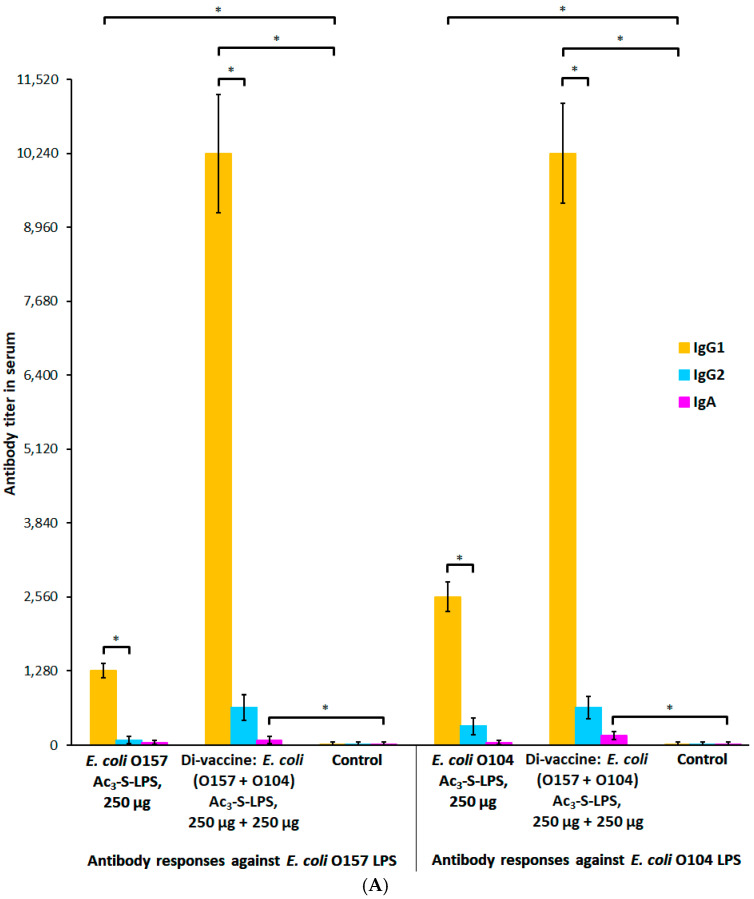
Systemic and mucosal humoral immune responses in mice after triple immunization with high dose of *E. coli* (O157 + O104) Ac_3_-S-LPS di-vaccine and its individual components. (**A**) Titers of specific IgG subclasses in sera increased in mice immunized i.p. three times with 250 µg of each Ac_3_-S-LPS *E. coli* O157 or Ac_3_-S-LPS *E. coli* O104 antigen or with 500 µg (250 µg + 250 µg) of di-vaccine. IgG1 was the dominant subclass in the sera. Differences between IgG1 and IgG2 titer increases were significant (*p* < 0.05, ANOVA). Ac_3_-S-LPS preparations induced 128-fold and higher rises in serum O-specific IgG1-titers, and these rises were significant (*p* < 0.05, ANOVA). Only di-vaccine elicited IgA response for both antigens in the sera and the increases in IgA titers over those in control sera were significant (*p* < 0.05, ANOVA). (**B**) Titers of specific IgG subclasses in the feces increased in mice immunized i.p. three times with 250 µg of each Ac_3_-S-LPS *E. coli* O157 or Ac3-S-LPS *E. coli* O104 antigen or with 500 µg (250 µg + 250 µg) of di-vaccine. IgG1 was the dominant subclass in the feces. Differences between IgG1 and IgG2 titer increases were significant only for di-vaccine (*p* < 0.05, ANOVA). Di-vaccine induced 64-fold rises in fecal O-specific IgG1-titers for both antigens and these rises were significant. Di-vaccine also elicited IgA response for both antigens in the feces, and the 8-fold increases in IgA titers over those in control feces were significant. The results shown are means ± SD. Significant difference between values are indicated (* *p* < 0.05, ANOVA).

**Table 1 vaccines-10-01854-t001:** *E. coli* (O157 + O104) Ac_3_-S-LPS di-vaccine and its individual components protect mice against intraperitoneal challenge with homologous STEC strains. BALB/c mice (10 per group) were i.p. immunized with 25 µg of each Ac_3_-S-LPS *E. coli* O157 or *E. coli* O104 antigen or with 50 µg (25 µg + 25 µg) of di-vaccine. LD_100_ was 2.6 × 10^7^ CFU/mouse for *E. coli* O157 and 1 × 10^8^ CFU/mouse for *E. coli* O104.

Preparation	Immunization Scheme			Infection on Day 45, Strain of *E. coli*	Survival Rate (%)
	1st Injection	2nd Injection	3rd Injection		
*E. coli* O157 Ac_3_-S-LPS	Day 0	Day 30	-	O157	60
	Day 0	Day 10	Day 20	O157	70
*E. coli* O104 Ac_3_-S-LPS	Day 0	Day 30	-	O104	60
	Day 0	Day 10	Day 20	O104	100
*E. coli* (O157+O104) Ac_3_-S-LPS	Day 0	Day 30	-	O157	100
	Day 0	Day 10	Day 20	O104	100
Control (sterile saline)	Day 0	-	-	O157	0
	Day 0	-	-	O104	0

**Table 2 vaccines-10-01854-t002:** *E. coli* (O157 + O104) Ac_3_-S-LPS di-vaccine and its individual components reduce bacterial counts in the feces of mice intragastrically inoculated with homologous STEC strains. BALB/c mice (10 per group) were immunized i.p. three times with 10-day intervals with 250 µg of each Ac_3_-S-LPS *E. coli* O157 or *E. coli* O104 antigen or with 500 µg (250 µg + 250 µg) of di-vaccine and intragastrically challenged with 1 × 10^9^ CFU/mouse of homologous STEC strains resistant to streptomycin (*E. coli* O157^Str^ and *E. coli* O104^Str^). The decreases in bacterial counts in fecal samples of triple immunized mice compared to control group were statistically significant (* *p* < 0.05, ANOVA).

Preparation	Infection on Day 45, Strain of *E. coli*^Str^	*E. coli* O157 or O104 Fecal Cell Count (lg CFU/g), Mean ± SD				
		2nd Day	5th Day	7th Day	9th Day	12th Day
*E. coli* O157 Ac_3_-S-LPS	O157	8.83 ± 0.03 *	8.52 ± 0.02 *	8.48 ± 0.04 *	8.34 ± 0.02 *	8.16 ± 0.01 *
*E. coli* O104 Ac_3_-S-LPS	O104	9.13 ± 0.02 *	8.98 ± 0.04 *	9.00 ± 0.01 *	8.79 ± 0.02 *	8.57 ± 0.03 *
*E. coli* (O157+O104) Ac_3_-S-LPS	O157	8.60 ± 0.01 *	8.51 ± 0.01 *	8.29 ±0.01 *	8.09 ± 0.03 *	8.04 ± 0.03 *
	O104	9.3 ± 0.01 *	9.0 ± 0.02 *	8.6 ±0.03 *	8.4 ± 0.04 *	8.2 ± 0.02 *
Control (sterile saline)	O157	9.55 ± 0.02	9.33 ± 0.03	9.27 ± 0.02	8.99 ± 0.05	8.81 ± 0.01
	O104	9.8 ± 0.04	9.5 ± 0.03	9.5± 0.02	9.0 ± 0.01	8.9 ± 0.04

## Data Availability

The data presented in this study are available on request from the corresponding author.

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
