# Peer review of "Molecular Lipopolysaccharide Di-Vaccine Protects from Shiga-Toxin Producing Epidemic Strains of Escherichia coli O157:H7 and O104:H4"

_vaccines, 2022, doi:10.3390/vaccines10111854_

Round 1

Reviewer 1 Report

The manuscript “Molecular Lipopolysaccharide Di-vaccine Protects from Shiga- 2 toxin Producing Epidemic Strains of Escherichia coli O157:H7 3 and O104:H4” by Dyatlov et al. describes the immunization with tri-acylated LPS antigen induces igG1 that protects them against the lethal doses of highly virulent human STEC strains, E. coli O157:H7 and E. coli 221 O104:H4, in murine model.

Please re-write the following sentence. Use “development” instead of “creation” word in the sentence, and the sentence gives an impression that the requirement of development of broad-spectrum vaccine is a demand of certain researchers and it may not be required in reality, which is not be the case in fact. So, please re-write it.

Line 228-229: The creation of a broad-spectrum E. coli vaccine is an urgent problem in the opinion of a number of researchers.

Author Response

We thank the Reviewer for supporting the publication of our study. We greatly appreciate his valuable comment, which helped us to improve the quality of our manuscript.

We rewrote the sentence on Line 228-229: The creation of a broad-spectrum E. coli vaccine is an urgent problem in the opinion of a number of researchers.

Now it reads: According to the opinion of experts in the field of enteric infections, the development of a broad-spectrum E. coli vaccine is highly desirable.

Reviewer 2 Report

The manuscript entitled “Molecular Lipopolysaccharide Di-vaccine Protects from Shiga-toxin Producing Epidemic Strains of Escherichia coli O157:H7 and O104:H4” is well-structured and well-written by the authors, who have edited each section in a careful and scientifically rigorous manner.

The 'Introduction' section is balanced and accompanied by up-to-date and relevant references.

The 'Results' section is clear and sufficiently detailed throughout, including the tables.

Regarding the 'Materials and Methods' section, there is an error in writing the temperature on line 314 where there is an excessive space between the number and the centigrade symbol and where the latter is not well written. Line 316 should specify the sterility of the saline solution and (I assume) distilled water used. However, the sterility of the reagents should always be specified and in any case guaranteed wherever they are indicated. I realise this is an obvious thing, but it is equally important to specify it clearly.

Nothing can be objected to the thorough discussion of the results and the clear and comprehensive illustrations offered by the authors. The only problem is the definition of Figures 2 and 3, which would be worth enlarging in order to interpret them more clearly and increase their definition.

The authors end their discussion with concluding remarks about their excellent results, but do not structure a real 'conclusion' that could be effective.

Finally, I believe that, after these minor corrections, the manuscript is absolutely worth publishing due to the relevance of the information it contains and the topicality of the research conducted.

Author Response

We thank Reviewer#2 for recommendation of our study for publication.  We greatly appreciate his valuable comments and critiques, which helped us to improve the quality of our manuscript.

We corrected temperature degree symbols as they became distorted in multiple places.

On Line 310 we added information about sterility of solutions: Wet cells were subsequently washed with filter-sterilized saline and distilled water, and resuspended in sterile water.

We retrieved Hi-res pictures of 1H NMR and would like to use them as supplemental figures (please see the attachments. Also we have lager tiff files). We do not have better Mass spectra images. However, in our opinion, their resolution is good enough to enlarge and analyze them.

We split the paragraph at  the very end of the Discussion section to better highlight the main conclusion about clinical applicability of our vaccine candidate.

Reviewer 3 Report

The authors report an interesting approach of using modified LPS as vaccine candidate for protection against shiga-toxin producing epidemic strains of E. coli. The manuscript requires some revision before its acceptance for publication.

General comment:

In several instances, articles (e.g. a, the) are missing.

Abstract

Line 14, … strains cause food-borne illnesses …

Line 16, … as well as in elderly.

Lines 22-23, the half sentence in parenthesis should be moved to the end of the sentence.

Introduction

Line 37, … causes various severe infections …

Line 49, have not yet been developed.

Line 73, LPS for clinical use.

Line 86, meaning of unlike?

Results

Lines 106-107, can this be detected from NMR spectra?

Line 136, Table 1, two antigens with 60 and 70 versus 60 and 100 % survival rate, how can this difference be explained?

Lien 151, 2 w/w % each?

Line 162, 4-fold?

Line 164, 256-fold?

Line 173, why is the described data not shown?

Discussion

Lines 278 and 282, meaning of “transudation”?

Materials and methods

Line 316, how was the sample lyophilised?

Author Response

We thank the Reviewer#1 for thoroughly reviewing our manuscript.

We greatly appreciate his valuable comments and suggestions, which helped us to improve the quality of our manuscript.

We carefully edited the manuscript and corrected errors in English language. Also we made specific corrections requested by the Reviewer.

To address the question about NMR spectra (Lines 106-107) we added two supplemental figures with peaks for rare sugars marked (please see the attachment).

Line 136, Table 1, two antigens with 60 and 70 versus 60 and 100 % survival rate, how can this difference be explained?

            This difference can be explained by the fact that antigens (Ac3-S-LPS) of different STEC-strains - E. coli O157 and E. coli O104 obviously have different protective potential, and the preparation efficacy depended on the immunization scheme.  The data are consistent with IgG serum antibody response:  the third injection of preparations elicited increases over those following the second injection and Ac3-S-LPS E. coli O104 elicited 2-fold IgG increases over those following the immunization with Ac3-S-LPS E. coli O157.

Line 173, why is the described data not shown?

            Considering that there is no protective effect of E. coli O157 and O104 Ac3-S-LPS at a dose of 25 μg against intragastric infection with homologous STEC strains, we decided not to insert this data in order not to “overload” the article with data. At the end of the paragraph we added a sentence to better highlight the absence of the effect for 25 μg dose: Therefore, 25 µg dose of E. coli O157 and O104 Ac3-S-LPS provided no protective effect.

Lines 278 and 282, meaning of “transudation” -  passage of serum or other body fluid through a membrane or tissue surface

Line 316, how was the sample lyophilised? We added the following information to the text: Wet cells were subsequently washed with filter-sterilized saline and distilled water, and resuspended in sterile water.  Then 200 mL aliquots were loaded into 1200 ml cylindrical glass vials (Fast-Freeze, Labconco, USA) and frozen in a freezer (Shell Freezer, Labconco, USA) at - 40 °C for 40 min. Then vials were transferred to a lyophilizer (FreeZone, Labconco, USA) and freeze-dried at -50 °C for 12 h.
